# *Helicobacter pylori* Infection: Antibiotic Resistance and Solutions for Effective Management in Africa

**DOI:** 10.3390/antibiotics12060969

**Published:** 2023-05-26

**Authors:** Mashiko Setshedi, Stella I. Smith

**Affiliations:** 1Division of Gastroenterology, Department of Medicine, University of Cape Town, Cape Town 7925, South Africa; 2Molecular Biology and Biotechnology Department, Nigerian Institute of Medical Research, Yaba, Lagos 100001, Nigeria; stellaismith@yahoo.com

**Keywords:** *Helicobacter pylori*, antibiotics, antimicrobial resistance, Africa

## Abstract

*Helicobacter pylori* (*H. pylori*) infection is ubiquitous worldwide, with prevalence rates of greater than 70% in Africa. Symptomatic patients present with foregut gastrointestinal symptoms which can be readily diagnosed with standardized non-invasive or invasive tests. The biggest challenge, however, is in the management of this condition with rising antimicrobial resistance rates to most of the antibiotics recommended for therapy. This is a problem worldwide, but more specifically in Africa, where the socio-economic and political climate is such that eradication of this organism seems impossible. Furthermore, the recommended antimicrobial susceptibility testing for drug resistance is not widely available in Africa due to the lack of infrastructural as well as human resources. With the widespread unregulated use of antibiotics in some parts of Africa, the figures of antimicrobial resistance are likely to soar. In the face of these significant challenges, this ‘perspectives’ article aims to address the issue of antimicrobial resistance in Africa, by providing achievable and targeted goals to curb the spread of infection and rising antimicrobial resistance.

## 1. Introduction

*Helicobacter pylori* (*H. pylori*) is the most common infection in the world, with 4.4 billion people infected [1]. Although data on *H. pylori* prevalence in Africa is limited, two recent meta-analyses reported prevalences of at least 70%, representing the highest pooled rates worldwide [1,2]. Although the prevalence of *H. pylori* has declined in developed nations [1,3], it is likely that it will remain high in Africa. This is due to the fact that transmission is related to poor sanitation [4] and low socio-economic status [5]. This makes H. pylori infection a condition of high priority in the continent. 

*H. pylori* has an affinity for the gastric mucosa, causing universal inflammation [6]. Although most patients are asymptomatic, in 10–20%, *H. pylori* is associated with gastritis, duodenitis, peptic ulcer disease [1,7] and various other extra-intestinal manifestations [8]. Its most feared complication is that of gastric adenocarcinoma [9], which has a dismal prognosis accounting for 9% of all cancer-related mortality [10,11]. Given the inflammation and clinical consequences, the Kyoto consensus and the latest Maastricht VI/Florence consensus report recommend that all infected patients should be treated [12,13]. Whilst this recommendation seems obvious, the management of *H. pylori* is complex and fraught with challenges in developed nations [14], but arguably more so in Africa. To illustrate this, some of these unique challenges in implementing the recommendations of the Maastricht VI/Florence consensus report in the African setting have been highlighted [15]. This has serious implications for our ability to adequately manage *H. pylori*.

Unquestionably, however, the most concerning and challenging issue in the management of *H. pylori* is the alarming rates of widespread antimicrobial resistance to most of the recommended antibiotics that form the backbone of *H. pylori* therapy [16,17]. The factors contributing to rising antimicrobial resistance are multifactorial, not least of all that in many parts of Africa, the regulation of drug prescriptions is not strict. In Africa, of 36.2% of individuals who receive antibiotics for their symptoms, a third do not receive the prescription from a doctor, whilst another third acquire them from informal/unregistered dispensing establishments [18]. Furthermore, there is lack of infrastructure and human resources to perform antimicrobial susceptibility testing (AST). This indiscriminate use of antibiotics and lack of AST laboratories further places Africa at a looming crisis of antimicrobial resistance that may be so severe as to place the control of *H. pylori* in Africa under serious threat. Additionally, the lack of AST facilities contributes to the practice of empiric antimicrobial therapy even after failed second line therapy. Therefore, with competing priorities of eradicating *H. pylori* vis-à-vis observing prudent prescribing practices and choosing effective antibiotics to reduce the rate of antibiotic resistance, quo vadis? How do we in Africa move forward? This article aims to interrogate the recommended therapies for *H. pylori* infection and antimicrobial resistance including AST in terms of its relevance and applicability in Africa, and to suggest practical and achievable approaches to curb this infection. It is hoped that these recommendations will afford clinicians in Africa some hope of adequately managing *H. pylori* to prevent complications.

## 2. Recommended Therapies for *H. pylori*

Several international guidelines from across the globe have been published to guide the diagnosis and management of *H. pylori* [13,19,20]. Notwithstanding some minor nuances, these guidelines are similar in many respects. Even the Egyptian guideline (the only guideline from Africa) [21], is strongly informed by these international guidelines. The Maastricht VI/Florence consensus report is the most recently published of these and recommends empiric bismuth quadruple therapy (BQT) or concomitant non-bismuth quadruple (NBQT) if individual AST is not available and clarithromycin resistance is >15% or unknown. In areas of low clarithromycin resistance (<15%), clarithromycin-based triple therapy (CTT) or BQT is acceptable as first-line therapy, if proven to be locally effective [13]. First-line therapies have eradication rates of more than 98% globally [22]. In Africa, in a meta-analysis of 22 studies from nine African countries, the pooled eradication rate was 79% [23]. This study had very high heterogeneity, with variable eradication rates depending on study design, duration of treatment, and test used to assess eradication. Despite this limitation, it represents one of the few studies reporting on *H. pylori* eradication in Africa and highlights the need for better-designed observational or randomized control trials of available therapies to determine best practices in each region. Second-line therapy is determined by prior choices for first-line therapy, drug availability and antimicrobial resistance. If BQT fails, empiric flouroquinolone (triple or quadruple) therapy or high-dose PPI-amoxicillin therapy is recommended, assuming low flouroquinolone resistance [13]. If CTT or NBQT first-line therapy fails, the options are BQT, flouroquinolone or high-dose PPI-amoxicillin therapy. Third-line therapy includes bismuth-containing regimens with a different combination of antibiotics, high-dose PPI-amoxicillin (if not used previously), or a rifabutin-containing rescue regimen. For all therapies, 14-day regimens are recommended over 7- or 10- day regimens, due to superior eradication rates [13]. Although in previous guidelines clarithromycin-based triple therapy was the treatment of choice, this therapy is no longer recommended as first-line empiric therapy [13]. Mounting evidence shows inferior eradication rates (<80%) with clarithromycin when used empirically [16]. Moreover, a Korean study confirmed not just the superiority of AST-based first-line CTT over empiric therapy, but also cost-effectiveness, [24] lending further credence to a sway from empiric CTT when resistance rates are high or unknown. Finally, AST-guided therapy also has utility before first-line therapy where antibiotic resistance to two antibiotics exists [25]. In terms of second-line therapies, levofloxacin-based therapy is also problematic due to the increasing reported resistance [16,26], although fourth-generation quinolones such as sitafloxacin in combination with a potassium competitive acid blocker (PCAB) may mitigate this [27]. However, the latter regimen has not been validated and is therefore not recommended outside of Japan [13]. Rifabutin has an excellent resistance profile and is highly effective [28], hence it is widely recommended as rescue therapy. This antibiotic, however, is not without risk in Africa, with high numbers of people living with the human immunodeficiency virus. This is because rifabutin is used for the treatment of mycobacterium avium-complex (MAC) and rifampicin-resistant mycobacterium tuberculosis. 

## 3. Antibiotic Resistance in Africa

The landscape of *H. pylori* management has been evolving since the discovery of the organism, and indeed continues to evolve. One of the most striking and concerning shifts over the last decade has been increasing antimicrobial resistance [16]. Clarithromycin, metronidazole and tetracycline resistance rates have almost doubled from 2010 to 2015 [29]. Globally, clarithromycin resistance was 34%; metronidazole resistance, 55%; and levofloxacin resistance, 24% in the World Health Organization (WHO) regions in 2018 [16], (Table 1). This is concerning as most of these antibiotics form the backbone of recommended first- or second-line therapies. This is notable in that the recommendations for antibiotic therapy in many of the guidelines are strongly influenced by astronomic increases in antimicrobial resistance [16]. Consequently, most guidelines have moved to recommending BQT or NQBT as first-line therapy. Antibiotic resistance is the most common cause for treatment failure in *H. pylori* [30,31]. In fact, resistance to clarithromycin is associated with a seven-fold risk of treatment failure when using a clarithromycin-based regimen [16]. *H. pylori* antimicrobial resistance is currently so alarming that in 2018, the WHO listed clarithromycin-resistant *H. pylori* as a high-priority organism for the development of new antibiotics [32]. 

The implications of increasing antimicrobial resistance for Africa may be even more serious. Although data on antimicrobial resistance is limited, in a meta-analysis reviewing antimicrobial resistance in Africa, clarithromycin resistance was 15%; metronidazole resistance, 91%; levofloxacin resistance, 14%; amoxycillin resistance, 38%; and tetracycline resistance, 13% [16]. This meta-analysis, however, is limited by the fact that only three countries from Africa (Cameroon, Congo and Senegal) were represented; thus, this data cannot be deemed as representative of Africa. Furthermore, resistance patterns are likely to vary by geographic region based on patient characteristics and prevailing prescribing practices. Nonetheless, these figures are extremely concerning because in many parts of Africa, bismuth is generally not used or is unavailable for reasons that are unclear. In South Africa, bismuth is no longer manufactured or imported since the COVID-19 epidemic. In a second meta-analysis that was more representative, including 14 countries in Africa (Nigeria, South Africa, Egypt, Kenya, Senegal, Ethiopia, Tunisia, Malawi, Congo-Brazzaville, Gambia, Uganda, Morocco, Algeria and Cameroon), antimicrobial resistance to commonly used agents was as follows: clarithromycin, 29.2%; metronidazole, 75.8%; levofloxacin, 17.4%; amoxycillin, 72.6%; and tetracycline, 48.7% [33] (Figure 1). This meta-analysis showed higher resistance rates for all the antibiotics except metronidazole, compared with the paper by Savoldi et al. Although both of these meta-analyses were published in the same year, the time period of assessment was longer in the study by Jaka et al. [33] (1986–2017), compared with the one by Savoldi et al. [16] (2007–2017). Therefore, the differences may be explained by the longer duration of observation and greater representation of countries in the former. In order to confirm these findings, more studies are needed, ideally a registry that includes all regions, that will collect the same variables, thus decreasing the heterogeneity associated with most meta-analyses using studies of different study designs. This, in turn, will allow for pooling of data that can be correctly interpreted to uncover true prevalences of antimicrobial resistance in Africa. This data will enable the development of relevant and useful therapy recommendations for *H. pylori* management in Africa. Traditionally, *H. pylori* treatment has been empiric, based on likely effectiveness of the antimicrobial eradication rates and possibly other factors such as the availability of these drugs. It is notable, however, that in the last decade, the focus has been moving more and more towards tailored therapy with AST, even for first-line therapy choice [13]. This approach is becoming more possible with the rise in availability of AST in the USA [34]. However, for phenotypic or genotypic AST, laboratories that are equipped with the required tools and expertise are needed. These are not widely available in Africa.

## 4. Comprehensive Management Solutions for Africa

### 4.1. Treat Symptomatic Patients

The comprehensive management of *H. pylori* and effective control of infection is difficult even in first-world settings. However, some gains are being made, given the declining prevalence rates of infection in developed nations. This is likely because of concerted efforts to accurately diagnose and appropriately treat the infection with effective antimicrobials. In Africa, where resources are scarce and systemic historical challenges are entrenched, the management of this infection poses even more of a challenge. For this reason, our approach should be practical, and targeting ‘low-hanging fruit’, with a long-term lens of eradicating this organism. Specifically, we should concentrate on targeting preventative measures to control the spread of infection. At the patient level, this involves making an accurate diagnosis in symptomatic patients, using available resources. All symptomatic patients should be treated as soon as the diagnosis is confirmed, with the best first-line therapy available, based on published data. Where no data exists, the most likely effective therapy should be used. Where bismuth is not available, clarithromycin may be the best choice for first-line therapy in many regions of Africa. This is based on the fact that clarithromycin has lower resistance rates than metronidazole, amoxycillin and tetracycline. Levofloxacin may be a reasonable option for second-line therapy. Where therapy has failed, other causes of failure must be interrogated and addressed; for instance, non-completion of therapy and/or adverse effects. The recommendation of PCABs, newer quinolones and other combinations is impractical in our setting, as these are not available and have not yet been validated. Treatment should be for 14 days, and patients must be adequately advised (in a language they best understand) of the importance of completing the course of therapy and be warned of potential side effects. In every case, it cannot be overemphasized that patients must be reviewed for symptoms and confirmation of eradication after at least 30 days post-therapy. In Africa, due to the lack of non-invasive testing in some areas, confirmation of eradication is not always done. This poses significant challenges because of the risk of potentiating antimicrobial resistance where treatment success or failure is not documented.

### 4.2. Centralized Antimicrobial Susceptibility Testing Facilities

If second-line treatment fails, AST is recommended. As demonstrated, available data in Africa is sparse, and thus not representative. Even when considering available data, there is wide variability in practice, antibiotic resistance rates and availability of antibiotics. Therefore, the most rational and practical approach is to standardize *H. pylori* diagnosis and treatment within geographic locations. This may involve sending specimens outside of the region of the treating hospital to neighbouring regions where these facilities exist. To facilitate this, treating clinicians, through their gastroenterology societies, need to reach out to scientists and microbiologists through publications, research platforms and even social media platforms, to enhance awareness. To this end, a group of gastroenterologists and scientists from eight African countries formed the African Helicobacter and Microbiota Group in 2021 [35], akin to the European group. The mandate of the group is to consolidate all aspects of *H. pylori* management, including research activities, in Africa. This will facilitate collaborations, so that silos are broken down. It is through these efforts that we can consolidate our efforts and reduce redundancy, such that we can collectively apply for funding that will facilitate the setting up of patient and biorepository registries, as well as identify existing laboratories as centralized facilities for the performance of AST. These laboratories can be capacitated to provide training to newly identified laboratories so that this work can be expanded. The aim is to have at least one AST laboratory in each region of Africa (Northern, Western, Eastern, Central and Southern African regions). Through this centralized mechanism, using unified protocols, over time, we will be able to accurately report antimicrobial resistance rates across Africa, both between and within regions. In parallel, another practical and short-term strategy would be for Africa to exploit the infrastructure borne out of the COVID-19 pandemic by performing real-time PCR. In this context, it is also possible that *H. pylori* reagents can be accessed at a cheaper price for AST since the pandemic. This strategy in the long-term will be fortified by a systematic tracking and reporting of practices (epidemiology, diagnostic tools used, treatment regimens prescribed), *H. pylori* eradication rates and related outcomes. This information would be a powerful tool in the fight against antimicrobial resistance in that it will bring to the forefront the possibility of an African *H. pylori* guideline that is context-specific, but, more importantly, evidence-based. These efforts combined will increase the likelihood of successful eradication of this infection. 

### 4.3. Screening of Symptomatic Family Members

While it is not practical to recommend population-wide screening, or screening of all family members living in the same household as patients, it is not unreasonable to suggest screening for those with symptoms within families of patients. This could simply involve a screen for symptoms, and, if present, using a non-invasive tool for diagnosis. In Africa, mainly a stool antigen test, and in a few centres, a UBT, are used for non-invasive testing for *H. pylori* [36]. Family screening (other than for gastric cancer in high-risk populations) is not recommended in the current guidelines, but given the high prevalence of infection in Africa, screening symptomatic family members for infection may be a feasible solution to preventing spread of the infection. The cost-effectiveness of this strategy however, would need to be determined.

### 4.4. Stakeholder Involvement

There are many stakeholders in the control of this infection. The burden of infectious disease over decades in Africa has been due to HIV, TB and malaria, with justified focus on these. Meanwhile, infections such as hepatitis B and *H. pylori* have been ignored. Both conditions currently have the least public-health-initiative footprint; and yet have very serious complications. The time has come to put *H. pylori* on the agenda as a serious infection. We need to engage politicians, policy makers, medical regulatory bodies, drug companies, research funding organizations and community leaders to increase awareness around *H. pylori* infection. We need educational campaigns, not just for healthcare providers, but for patients, too, so that we can collectively take ownership of infection control, e.g., simple measures such as handwashing, improved hygiene and responsible use of antibiotics. Other risk-reducing practices, such as avoiding non-steroidal anti-inflammatory drugs (NSAIDS), testing and treating for *H. pylori* in patients on long-term aspirin, anticoagulants, steroids, etc., and reducing alcohol intake and smoking, are important. These need to be included in the comprehensive management of this disorder. At a wider scale, all governments, through bodies such as the African Union (a continental union consisting of 55 member states in Africa), can be made conscious of the need for improved infrastructure in Africa, to improve the general living conditions of people. This will improve control of the increasing antimicrobial resistance, not just for *H. pylori*, but for many other bacterial infections that are so common in Africa. Unfortunately, at this stage, neither a vaccine nor probiotics are available in clinical practice, and thus these cannot be included in the armamentarium in the preventative fight against *H. pylori*.

## 5. Conclusions

*H. pylori* infection is not only the most common infection worldwide, but is a group 1 carcinogen. In 20% of patients with symptomatic infection, it is associated with significant morbidity and mortality. Although the diagnosis of *H. pylori* is standardized and relatively easy to make, treatment and eradication of the organism is fraught with significant challenges, the most concerning being antimicrobial resistance. Although AST facilities are becoming more commonplace in developed nations, this is certainly not the case in Africa, and with the injudicious use of antibiotics, antimicrobial resistance is at crisis point. The immediate priority should be to effectively treat those infected, scale up preventative measures, and involve patients, the medical and scientific professionals, as well as other key stakeholders to combine efforts to successfully eradicate *H. pylori*. There has been a constant evolution in our understanding of *H. pylori*; as such, the usual strategies for managing bacterial infections are proving not to be sufficient. In this regard, novel approaches are needed to tackle *H. pylori*.

## Figures and Tables

**Figure 1 antibiotics-12-00969-f001:**
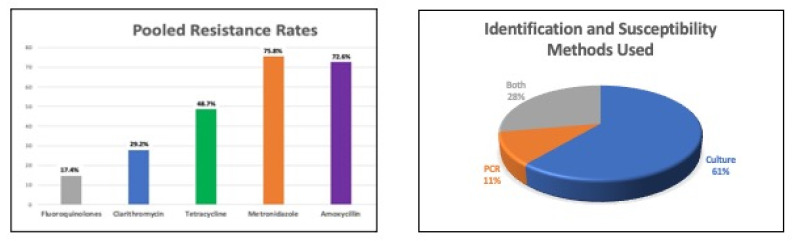
Antimicrobial resistance in Africa, adapted from Jaka 2016.

**Table 1 antibiotics-12-00969-t001:** Pooled primary antibiotic resistance by World Health Organization Region.

WHO Region	Clarithromycin, % (95% CI)	Metronidazole, % (95% CI)	Amoxycillin, % (95% CI)	Levofloxacin, % (95% CI)
Africa	15 (0–30)	91 (87–94)	38 (32–45)	14 (12–28)
Americas	10 (4–16)	23 (2–44)	10 (2–19)	15 (5–16)
Eastern Mediterranean	33 (23–44)	56 (46–66)	14 (8–20)	19 (0–29)
European	18 (16–20)	32 (27–36)	0 (0–0)	11 (9–13)
South-East Asia	10 (5–16)	51 (26–76)	2 (0–5)	30 (14–46)
Western Pacific	34 (30–38)	47 (37–57)	1 (1–1)	22 (17–28)
Overall	34 (30–38)	55 (51–59)	1 (1–1)	24 (21–26)

## Data Availability

Not applicable.

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
