# Peer review of "Helicobacter pylori Infection: Antibiotic Resistance and Solutions for Effective Management in Africa"

_antibiotics, 2023, doi:10.3390/antibiotics12060969_

Round 1

Reviewer 1 Report

The title of the study is not so meaningful, if possible change it like replace solutions with recommendations, suggestions etc. 

Please provided references for the statement 'Although data on H. pylori prevalence in Africa is limited, two recent meta-analysis reported prevalence of at least 70%, which represent the highest pooled rates worldwide'.

I would recommend to add possible antibiotics (like amoxicillin, metronidazole and clarithromycin etc) for the treatment of H pylori while describing ongoing antibiotics resistance related to them particularly in Africa.

'Rifabutin has an excellent resistance profile and is 94 highly effective' is not meaningful please change it to like it has good effectiveness with little or no resistance etc.

 These figures are extremely concerning particularly because in 115 many parts of Africa, bismuth is unavailable (line 115), can you please explain it about the non-availability of bismuth? please add this in text too

I can see double space placed in the text by the authors please carefully see them and correct where necessary

Overall, English language and grammar need modifications. Please go through the whole manuscript carefully, avoid using too long sentences, try to present the things in a simple and convincing way.

Author Response

  1. The title of the study is not so meaningful, if possible change it like replace solutions with recommendations, suggestions etc. Thank you for the comment. We considered changing it to "recommendations" as per your suggestion. In the end we decided against it because this article is a perspectives (which essentially are opinions about published data). For this reason we felt that "solutions" was a more appropriate word. We felt that "recommendations", would make more sense if this article was a guideline or consensus statement by multiple authors on the subject. Thus we elected to retain the word "solutions". Line 3.
  2. Please provided references for the statement 'Although data on H. pylori prevalence in Africa is limited, two recent meta-analysis reported prevalence of at least 70%, which represent the highest pooled rates worldwide'. These have been provided. Line 28.
  3. I would recommend to add possible antibiotics (like amoxicillin, metronidazole and clarithromycin etc) for the treatment of H pylori while describing ongoing antibiotics resistance related to them particularly in Africa. This is discussed under the heading "Treat symptomatic patients", understanding the limitations of suggesting metronidazole and amoxycillin given the high rates of resistance. Lines 170-174.
  4. Rifabutin has an excellent resistance profile and is highly effective' is not meaningful please change it to like it has good effectiveness with little or no resistance etc. We have added the phrase "and has an excellent resistance profile". Line 92-93.
  5. These figures are extremely concerning particularly because in 115 many parts of Africa, bismuth is unavailable (line 115), can you please explain it about the non-availability of bismuth? please add this in text too. This has been added. Lines 121-122.
  6. I can see double space placed in the text by the authors please carefully see them and correct where necessary. This has been checked and corrected.
  7. Overall, English language and grammar need modifications. Please go through the whole manuscript carefully, avoid using too long sentences, try to present the things in a simple and convincing way. The manuscript was sent to a first language English speaker who made changes. Long sentences were shortened, redundant words were removed. 

Reviewer 2 Report

The manuscript by Mashiko Setshedi and Stella I. Smith discusses the evolution of the management of Helicobacter pylori infections in Africa.
This article is generally well written, but deserves to be improved according to my recommendations below.
Global: prefer passive voice. put names of bacteria, and "e.g." in italics.
Conclusion: it could be interesting if the authors summarize (in a geographical figure for example) the prevalence and antibiotic resistance data found in the literature to give the non-African reader a better idea of the epidemiology of the continent.
In addition, it might be interesting to place these prevalence/resistance data in the global landscape, allowing for a better understanding of potential management challenges.
Family screening: it may be interesting to discuss the non-invasive screening modalities available in Africa.

Author Response

  1. Global: prefer passive voice. put names of bacteria, and "e.g." in italics. This has been done.
  2. Conclusion: it could be interesting if the authors summarize (in a geographical figure for example) the prevalence and antibiotic resistance data found in the literature to give the non-African reader a better idea of the epidemiology of the continent. We have added a table (depicting resistance rates in WHO regions including Africa).
  3. In addition, it might be interesting to place these prevalence/resistance data in the global landscape, allowing for a better understanding of potential management challenges. We have added a table (depicting resistance rates in WHO regions including Africa). In addition, we have created a figure of resistance in Africa for easier comparisons between other areas and Africa to be made.
  4. Family screening: it may be interesting to discuss the non-invasive screening modalities available in Africa. This has been added. Lines 217-218.

Reviewer 3 Report

This is a review about H. pylori infection: antibiotic resistance and solutions for effective management in Africa. But you haven't mentioned anywhere what kind of study it is. Please add this information to your article.

Lines 11 and 26... write Helicobacter pylori with italics.

Lines 27-28... you have no references for the two meta-analyses.

The part with solutions for effective management is treated a bit superficially. I recommend you to make improvements, to discuss this aspect in more detail.

The article does not present any table or figure. It is recommended to include at least two.

The article presents 35 references, being up to date. However, for a review, there are too few references. In my opinion, this article is too short. By extending it, you will be able to improve the number of references.

Author Response

  1. This is a review about H. pylori infection: antibiotic resistance and solutions for effective management in Africa. But you haven't mentioned anywhere what kind of study it is. Please add this information to your article. This article is a "perspectives". According to the journal this is the definition of this type of article: 

    Perspectives: Perspectives are opinion or commentary articles that express personal opinions about existing studies that have great impact on the antibiotics research community. Perspectives should have a main text of around 3500 words at minimum, with at least 20 references.

    The word "perspectives" was added to the very top of the article, before the title. Line 1.
  2. Lines 11 and 26... write Helicobacter pylori with italics. This has been done.
  3. Lines 27-28... you have no references for the two meta-analyses. The references have been added.
  4. The part with solutions for effective management is treated a bit superficially. I recommend you to make improvements, to discuss this aspect in more detail. This has been done.
  5. The article does not present any table or figure. It is recommended to include at least two. A table and figure have been added.
  6. The article presents 35 references, being up to date. However, for a review, there are too few references. In my opinion, this article is too short. By extending it, you will be able to improve the number of references. As stated in point 1, this article is a "perspectives" not a review article. Because the manuscript had some changes made (as suggested by the other reviewers), we added a 2more references.

Round 2

Reviewer 1 Report

Thank you to authors for considering my comments/suggestions

Reviewer 2 Report

The manuscript has been well-improved according to my previous comments.

Reviewer 3 Report

The authors answered all queries.